



# Investigation and amelioration of long-term instrumental drifts in water vapor and nitrous oxide measurements from the Aura Microwave Limb Sounder (MLS) and their implications for studies of variability and trends

Nathaniel J. Livesey[1], William G. Read[1], Lucien Froidevaux[1], Alyn Lambert[1], Michelle L. Santee[1], Michael J. Schwartz[1], Luis F. Millán[1], Robert F. Jarnot[1], Paul A. Wagner[1], Dale F. Hurst[2], Kaley A. Walker[3], Patrick E. Sheese[3], and Gerald E. Nedoluha[4]

[1]Jet Propulsion Laboratory, California Institute of Technology, Pasadena, California, USA.
[2]Cooperative Institute for Research in Environmental Sciences, University of Colorado, Boulder, Colorado, USA, and NOAA Global Monitoring Laboratory, Boulder, Colorado, USA.
[3]University of Toronto, Ontario, Canada.
[4]Remote Sensing Division, Naval Research Laboratory, Washington, DC, USA.

**Correspondence:** Nathaniel .J. Livesey (Nathaniel.J.Livesey@jpl.nasa.gov)

**Abstract.** The Microwave Limb Sounder (MLS), launched on NASA's Aura spacecraft in 2004, measures vertical profiles of the abundances of key atmospheric species from the upper troposphere to the mesosphere with daily near-global coverage. We review the first 15 years of the record of $H_2O$ and $N_2O$ measurements from the MLS 190-GHz subsystem (along with other 190-GHz information), with a focus on their long-term stability, largely based on comparisons with measurements from

other sensors. These comparisons generally show signs of an increasing drift in the MLS "version 4" (v4) $H_2O$ record starting around 2010. Specifically, comparisons with v4.1 measurements from the Atmospheric Chemistry Experiment – Fourier Transform Spectrometer (ACE-FTS) indicate a ~2–3%/decade drift over much of the stratosphere, increasing to as much as ~7%/decade around 46 hPa. Larger drifts, of around 7–11%/decade, are seen in comparisons to balloon-borne frost point hygrometer measurements in the lower stratosphere. In contrast, the MLS v4 $N_2O$ product is shown to be generally decreasing

over the same period (when an increase in stratospheric $N_2O$ is expected, reflecting a secular growth in emissions), with a more pronounced drift in the lower stratosphere than that found for $H_2O$. Detailed investigations of the behavior of the MLS 190-GHz subsystem reveal a drift in its "sideband fraction" (the relative sensitivity of the 190-GHz receiver to the two different parts of the microwave spectrum it observes). Our studies indicate that sideband fraction drift accounts for much of the observed changes in the MLS $H_2O$ product and some portion of the changes seen in $N_2O$. The 190-GHz sideband fraction drift

has been corrected in the new "version 5" MLS algorithms, which have now been used to reprocess the entire MLS record. As a result of this correction, the MLS v5 $H_2O$ record shows no statistically significant drifts compared to ACE-FTS. However, statistically significant drifts remain between MLS v5 and frost point measurements, though they are reduced. Drifts in v5 $N_2O$ are about half the size of those in v4 but remain statistically significant. Scientists are advised to use MLS v5 data in all future studies. Quantification of inter-regional and seasonal-to-annual changes in MLS $H_2O$ and $N_2O$ will not be affected by the drift.

However, caution is advised in studies using the MLS record to examine long-term (multi-year) variability and trends in either





of these species, especially $N_2O$; such studies should only be undertaken in consultation with the MLS team. Importantly, this drift does not affect any of the MLS observations made in other spectral regions such as $O_3$, HCl, CO, ClO, or temperature.

## 1   Introduction

The Microwave Limb Sounder (MLS, Waters et al., 2006) is one of four instruments launched on NASA's Aura mission
(Schoeberl et al., 2006) in July 2004 and has operated essentially continuously from launch to the present. The MLS GHz antenna looks in the "forward" direction from the Aura spacecraft and vertically scans Earth's limb from the surface to ~95 km altitude every ~26 s, yielding ~3500 scans per day. Aura is in a near-polar (98°-inclined) sun-synchronous orbit, enabling MLS atmospheric observations from 82°S–82°N each orbit. The instrument measures thermal microwave signals in the 118, 190, 240, 640, and 2500 GHz regions of the spectrum, with separate microwave receiver and spectrometer assemblies for each region
(two each for the 118 and 2500 GHz regions, and one for each of the others, for a total of seven receivers). From these radiance measurements, the "Level 2" retrieval algorithms (Livesey et al., 2006; Read et al., 2006; Schwartz et al., 2006) deduce vertical abundance profiles of $O_3$, $H_2O$, $N_2O$, CO, $HNO_3$, HCl, ClO, HOCl, $CH_3Cl$, BrO, OH, $HO_2$, HCN, $CH_3CN$, $CH_3OH$, and $SO_2$, along with temperature, geopotential height, cloud ice amount, and relative humidity.

With the Aura MLS data record now more than 16 years long (MLS having far exceeded its five-year design life), many
studies are employing it, sometimes in conjunction with other observational datasets, to characterize long-term atmospheric variability. Published investigations include the quantification of stratospheric ozone trends and their potential attribution to ozone layer recovery (e.g. Steinbrecht et al., 2017; Strahan and Douglass, 2018; Ball et al., 2018, 2019; Chipperfield et al., 2018; Petropavlovskikh et al., 2019), examination of trends in tropospheric ozone (e.g. Gaudel et al., 2018), and studies of long-term variability in stratospheric water vapor (e.g. Lossow et al., 2018) and long-lived tracers and halogenated gases
(e.g. Stolarski et al., 2018; Froidevaux et al., 2019). The MLS-observed interannual and long-term variability in trace gases has, in turn, been used to examine associated underlying variability in transport into, within, and out of the stratosphere (e.g., Dessler et al., 2014; Neu et al., 2014; Verstraeten et al., 2015; Han et al., 2019; Diallo et al., 2019; Ruiz et al., 2021). Accordingly, increased attention is being paid to the long-term stability of the MLS record, to ensure that all such studies are placed on a sound footing. In the case of the MLS ozone "standard" product (obtained from radiance measurements
by the MLS 240-GHz receiver), stability has previously been demonstrated through comparisons with ground-based lidar observations (Nair et al., 2012), and comparisons versus MLS data have been used as a measure of the stability of other long-term records (e.g. Adams et al., 2014; Eckert et al., 2014). Confidence in the stability of MLS ozone was further underscored in a very thorough study (Hubert et al., 2016) comparing ozone profiles from MLS and a range of other spaceborne sensors to a comprehensive set of coincident measurements from both ozone sondes and ground-based lidar. The authors concluded that
the MLS "version 3" ozone record (at least through the May 2013 date then considered) was stable to within ±1.5%/decade in the middle stratosphere, and ±2%/decade in the upper stratosphere, with these small drifts with respect to the lidar and sonde records not being statistically significant at 95%. Larger and statistically significant drifts compared to the lidar and sonde records were seen at lower altitudes, though they were all within ±5%/decade.





In contrast to ozone, we now have strong evidence that in "version 4" (v4) the MLS $H_2O$ and $N_2O$ products, both measured by the 190-GHz receiver, show larger and statistically significant drifts. This paper quantifies drifts in these and other species measured by the MLS 190-GHz receiver, discusses factors that may be giving rise to the drifts, describes efforts by the MLS team to ameliorate those factors, and provides updated guidance to users of the affected MLS products. These drifts were
first noted by Hurst et al. (2016) in comparisons of MLS $H_2O$ with measurements from balloon-borne NOAA Frost Point Hygrometer (FPH) and Cryogenic Frostpoint Hygrometer (CFH) instruments launched from Boulder, Colorado and other locations. The Boulder comparisons show indications of an approximately +10%/decade drift in MLS lower stratospheric water vapor starting around 2009/2010. Section 2 revisits these findings and discusses further analyses and comparisons with other sensors that also show evidence for a positive drift in MLS $H_2O$, albeit a slower one than that indicated by the frost
point record. Section 2 also quantifies a drift between the MLS standard ozone product, measured at 240 GHz, and a diagnostic ozone product derived from an ozone line measured by the MLS 190-GHz receiver. Section 2 finishes by examining MLS $N_2O$ measurements, which are drifting to smaller values. Section 3 provides background on the MLS 190-GHz subsystem and its measurements and describes insights obtained into the probable underlying contributors to the observed drifts. Section 3 also details how, in the new MLS "version 5" (v5) data record, much of the $H_2O$ drift has been ameliorated, and drifts in $N_2O$ and
190-GHz $O_3$ have been reduced. Finally, section 4 provides a summary and, most importantly, guidance for users of the MLS $H_2O$ and $N_2O$ datasets as to how the drifts should be factored into future studies using those data.

## 2   Observations and comparisons

This section examines observed drifts in the MLS $H_2O$, $N_2O$, and 190-GHz $O_3$ products. MLS 190-GHz signals are also the source of the MLS HCN product and contribute to the $HNO_3$ product. However, the behavior of the latter two species is better
considered as part of the discussion of the origin of the drifts in the first three products. Accordingly, consideration of the MLS HCN and $HNO_3$ products is deferred to section 3.

### 2.1   MLS water vapor observations

#### 2.1.1   Comparisons with the balloon-borne hygrometer record

In contrast with the lower and middle troposphere, for which a wealth of well-characterized observations exist to validate
spaceborne humidity sensors, in the upper troposphere and lower stratosphere correlative observations of water vapor are far less frequent and sparser. Although both operational radiosondes and spaceborne Global Navigation Satellite System-Radio Occultation sounders are able to profile temperature well into the stratosphere, neither provide scientifically useful water vapor information in or above the upper troposphere. We note that several in situ instruments have obtained lower stratospheric water vapor observations from high-altitude aircraft, and these observations have been used for validation of the MLS water
vapor product (e.g., Read et al., 2008; Weinstock et al., 2009). However, their campaign-based sampling is extremely sparse





both temporally and spatially, severely hampering their application to validation of long-term variability in global spaceborne observations.

The longest near-continuous systematic record of in situ stratospheric water vapor observations comes from balloon-borne frost point hygrometer instruments (Mastenbrook and Oltmans, 1983; Vömel et al., 2007, 2016; Hall et al., 2016; Hurst et al., 2011, 2014, 2016) launched from Boulder, Colorado. Additional frost point instruments have been routinely launched from a small number of locations, though not for as long a period. The typical uncertainty on frost point measurements of water vapor mixing ratios in the lower stratosphere is better than 6% (Hall et al., 2016; Vömel et al., 2016). Most of this uncertainty is in the form of "random" error due to oscillations in the feedback loop that maintains a stable layer of frost on a temperature-controlled mirror. Absolute accuracy is estimated to be significantly better. Specifically, through the careful calibration of each instrument, $<0.05\,\mathrm{K}$ inaccuracies in frost point temperatures result in systematic errors of $<1\%$ in stratospheric water vapor partial pressures. However, biases in radiosonde pressure measurements can increase mixing ratio biases to 4% or greater in the stratosphere (Hall et al., 2016).

Frost point sonde observations have underpinned the validation of stratospheric water vapor measurements from the MLS instruments (Vömel et al., 2007; Read et al., 2007) and other spaceborne sensors. As reported by Hurst et al. (2016), comparisons between the frost point record and MLS v4 $H_2O$ indicate a divergence between these two water vapor datasets in the lower stratosphere, commencing around 2010, with the MLS values increasing relative to the frost point record. Figure 1 presents an update of the Hurst et al. (2016) comparisons between MLS v4 and frost point measurements at Lauder (New Zealand), Hilo (Hawaii, USA), and Boulder (Colorado, USA). These timeseries have been extended with additional years and are shown with the sign convention switched compared to those presented by Hurst et al. (2016), in order to be consistent with later figures in this paper. As with the original study, the MLS and frost point data are quality screened using established rules described in Livesey et al. (2020) and Hurst et al. (2016), respectively, and the frost point profiles have been smoothed by the MLS "least squares operator" and averaging kernel.

The year 2010 was chosen as the starting point for the fitted drift in light of the findings of the Hurst et al. study, substantiated by the observed behavior of the MLS 190-GHz receiver subsystem as discussed in section 3. These and most other fits described in this paper are simple linear regressions of differences between individual MLS Level 2 profiles and coincident measurements. All differences carry equal weight in the regression (i.e., no accounting is made for estimated precision or other factors). The strongest drifts, of ~11%/decade, are seen in the 68 hPa comparisons for all three sites, with the Lauder and Boulder drifts having ~±3%/decade confidence, and the Hilo drift being less definitive, though still clearly statistically significant, with a confidence interval of ~±7%/decade. All three locations indicate a slower drift at 100 hPa than at the higher altitudes, particularly Hilo, for which the 22 hPa drift is also smaller than that at 68 and 46 hPa.

Except where noted otherwise, all of the statistical fits in this paper establish significance at 95% using a block bootstrap resampling of the fit residuals (e.g., Froidevaux et al., 2019, and references therein) that uses one-year blocks and allows for block replacement. For succinctness, rather than referring to the possibly asymmetric 95% confidence intervals from this analysis (which would require two numbers to describe), we will use $\pm2\sigma$ bounds to describe uncertainty in drifts, where the

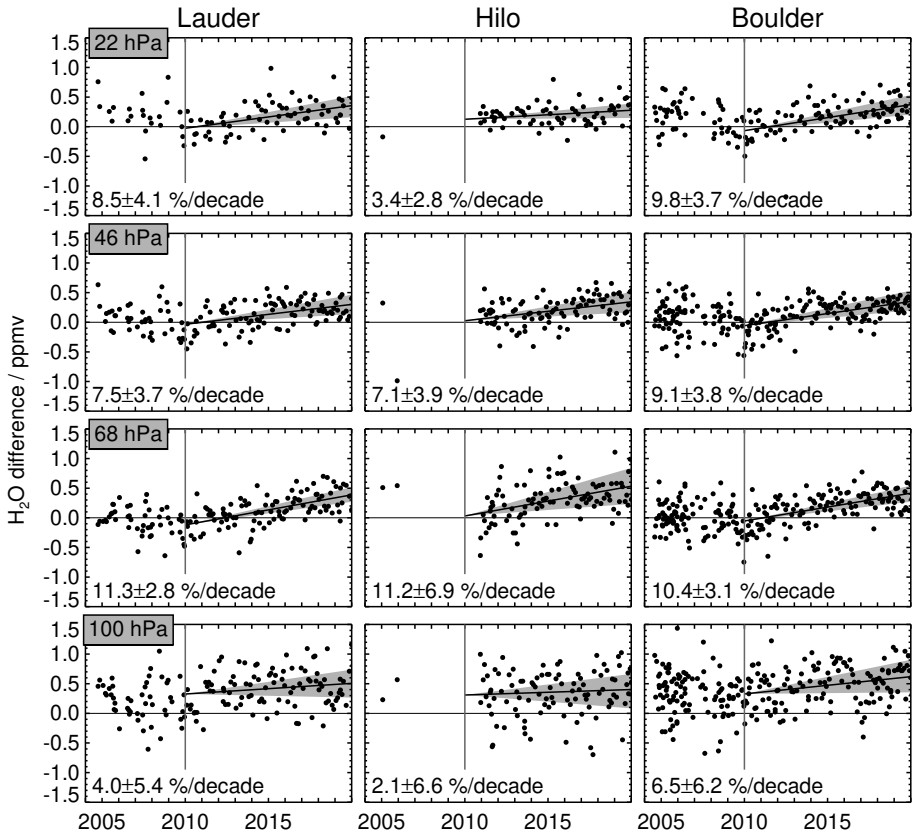

**Figure 1.** Timeseries of differences between MLS v4 and coincident frost point sonde measurements made from Lauder (45°S, 170°E, left), Hilo (20°N, 155°W, center) and Boulder (40°N, 105°W, right) at selected pressure levels (rows). Coincidence criteria are, as for Hurst et al. (2016), $\pm 2°$ latitude, $\pm 8°$ longitude, and $\pm 18$ hours. Positive differences imply that MLS measures greater humidity than the in situ observation. Black lines show a linear fit to the post-2010 differences. Grey shading shows the 95% confidence interval for the slope (from a block-bootstrap resampling of year-long blocks of the fit residuals). Fitted drift rates are quoted in each panel, with uncertainties corresponding to the $2\sigma$ spread of the bootstrap results. Comparisons at the intermediate vertical levels (not shown) indicate similar behavior.

standard deviations, $\sigma$, are those resulting from the bootstrapping analysis. The differences between the $2\sigma$ and 2.5%/97.5% thresholds are small in the cases shown here.

Given that each frost point sonde is an independently calibrated instrument, and the calibration scheme is referenced back to National Institute of Standards and Technology (NIST)-standard temperature sensors as well as to a small archive of previously calibrated mirror thermistors, it seems unlikely that this drift reflects any systematic evolution of the calibration of the frost point sonde record. That said, we note the findings of Lossow et al. (2018), particularly their Figure 8, which shows two multi-year periods during 1987–2001 when the frost point record at Boulder departed by 5–10% from both model-based and satellite-based measures of stratospheric water vapor. These 2- and 4-year periods of disparity may have resulted from atypically large, batch-dependent biases in radiosonde pressure sensors that propagated directly to the frost point mixing ratios. Such pressure





biases became easily correctable after Global Positioning System (GPS) receivers were added to radiosondes in 2001, and similar disparities were not observed again until ~2010, when MLS water vapor retrievals began to drift.

### 2.1.2 Comparisons with other spaceborne sensors

Another source of near-continuous long-term stratospheric water vapor observations is the spaceborne Atmospheric Chemistry
Experiment – Fourier Transform Spectrometer (ACE-FTS) on the Canadian SciSat-1 spacecraft, launched in 2003. ACE-FTS observes in a solar occultation geometry, providing vertical profiles of water vapor (and $N_2O$ and many other trace gases) each spacecraft sunset and sunrise, giving ~30 profiles of each measured species per day (Bernath et al., 2005). The SciSat-1 orbit was chosen to preferentially cover high latitudes, limiting the occasions on which tropical and mid-latitude regions are sampled to six or so one-week periods per year. Figure 2 shows drift rates inferred from linear fits to MLS v4 and ACE-FTS version 4.1
water vapor observations within various latitude bands for 2005–2010 (upper) and 2010–2019 (lower). ACE-FTS observations were screened according to the approach described in Sheese et al. (2015), and profiles are considered coincident if they are within $\pm 1°$ latitude, $\pm 8°$ longitude and $\pm 12$ hours. Given the similarity of the MLS and ACE-FTS vertical resolution, no averaging kernels were applied in this comparison.

The strongest post-2010 drifts between MLS v4 and ACE-FTS water vapor are around 4–7%/decade ($\pm 6$%/decade), seen
at the 68 and 56 hPa levels in the mid-latitudes. There are strong indications of a drift at these and other pressure levels in all the latitude regions shown, though at most levels it is not significant at $2\sigma$ in the 20°S–20°N band. The extratropical post-2010 drifts show a minimum value at 46 hPa, above which a fairly uniformly (and typically significant) positive drift of around 2–3%/decade is found. We note that these drifts are mostly smaller than those seen in the frost point comparisons (Figure 1), particularly at the higher altitudes (e.g., 22 hPa). By contrast, the 2005–2010 period shows essentially no statistically significant
drifts between MLS v4 and ACE-FTS (a few levels/latitudes have $> 2\sigma$ drifts, though no more than might be expected when using a ~95% confidence criterion for each of ~24 levels over three latitude bands).

Recently, the team responsible for the Sounding of the Atmosphere using Broadband Emission Radiometry (SABER) instrument on the Thermosphere Ionosphere Mesosphere Energetics and Dynamics (TIMED) mission, launched in 2002, has started providing a water vapor product from the lower stratosphere to the thermosphere (Rong et al., 2019). These data are
estimated to have 4% precision on an individual profile below 60 km and have been shown to agree with those from other sensors to within ~20% or better. Figure 3 shows estimated drifts between MLS v4 and SABER version 2.07 water vapor using the same coincidence criteria as used for ACE-FTS in Figure 2. In contrast with ACE-FTS and the frost point observations, comparisons between MLS v4 and SABER show no statistically significant drift except perhaps in a narrow region around 3 hPa, where a 2–3%/decade drift is seen in the southern and tropical latitude bands (a similar but not statistically significant
pattern is seen in the northern latitudes also). The $2\sigma$ uncertainties on the estimated lower stratospheric drifts are significantly larger than those for the corresponding ACE-FTS comparisons. In many cases the $2\sigma$ uncertainty in MLS versus SABER drift encompasses the drift estimated from the MLS and ACE-FTS comparisons.

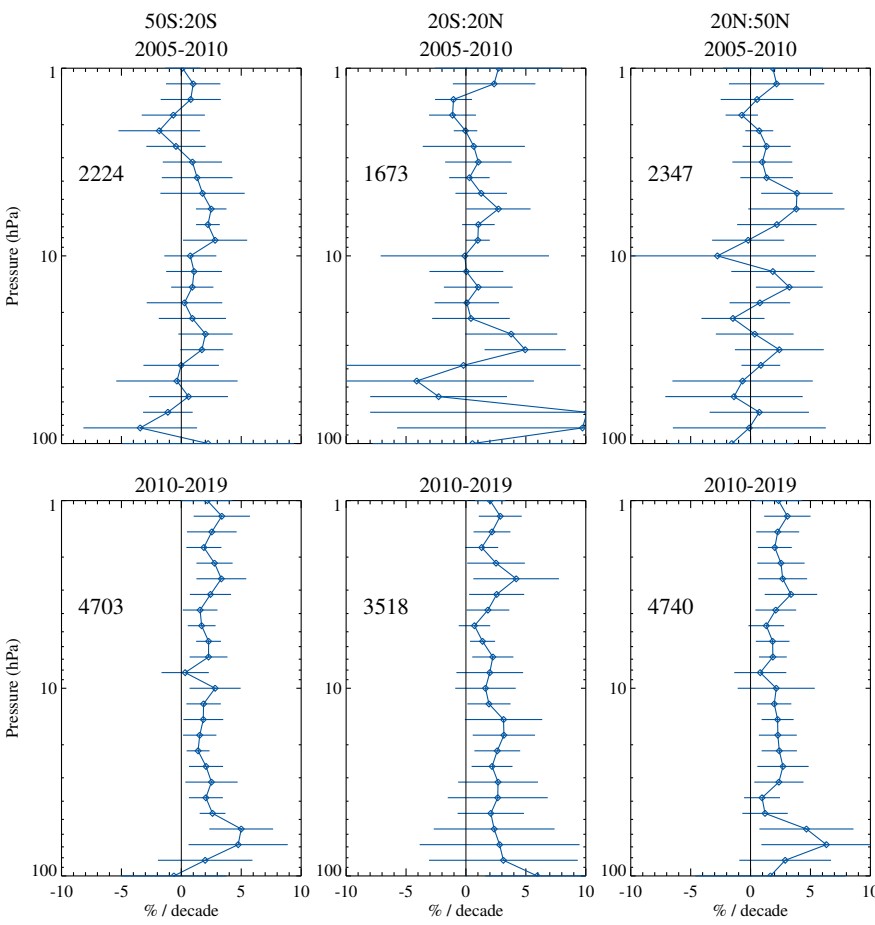

**Figure 2.** Drifts between MLS v4 and ACE-FTS version 4.1 observations of stratospheric water vapor for various latitude bands (columns) for 2005–2010 (upper) and 2010–2019 (lower). Numbers in each panel denote the total number of coincident (within $\pm 1°$ latitude, $\pm 8°$ longitude and $\pm 12$ hours) MLS/ACE observations, and error bars denote the $2\sigma$ block bootstrap interval for the linear fit term. Positive numbers imply that MLS values are drifting upwards compared to those from ACE-FTS.



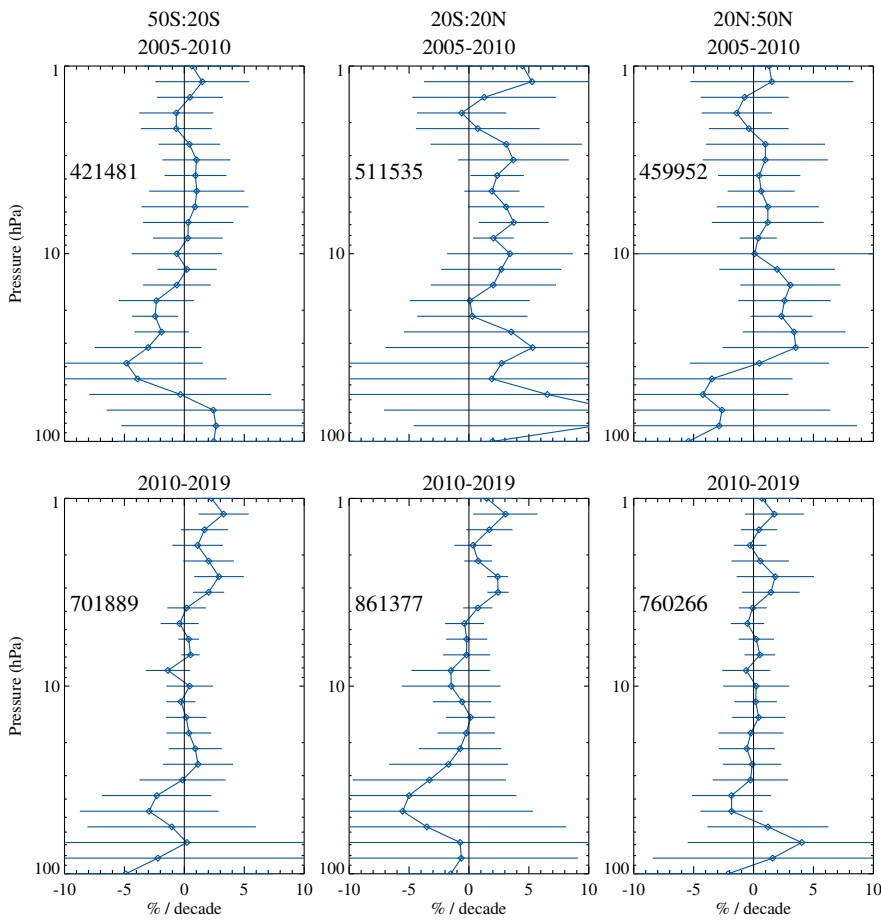

**Figure 3.** As Figure 2 but comparing MLS v4 to SABER version 2.07.

### 2.1.3 Comparisons with ground-based observations

Water vapor in the upper stratosphere and lower mesosphere can be measured from the ground by zenith-viewing microwave spectrometers. Figure 4 compares observations of water vapor at around 55 km altitude from MLS and the Water Vapor Millimeter-wave Spectrometer (WVMS) instruments (Nedoluha et al., 2011). The WVMS measurements are taken from Net-

5 work for the Detection of Atmospheric Composition Change (NDACC) sites at Table Mountain (California, USA, upper), Mauna Loa (Hawaii, USA, middle), and Lauder (New Zealand, lower). The MLS profiles have been smoothed using the WVMS averaging kernel. The WVMS retrievals use an MLS-based climatology, which includes seasonal variations (but not interannual variations, thus it is not affected by the MLS drift). These comparisons, from the altitude region where the WVMS measurements are strongest, give no reason to reject a null-hypothesis of zero drift between the two instruments. That said, in

10 the case of Mauna Loa and Lauder, the $2\sigma$ uncertainty encompasses the ~2%/decade drift in the upper stratosphere estimated from MLS and ACE-FTS comparisons. The $2\sigma$ uncertainty in the Table Mountain comparisons does not quite encompass such





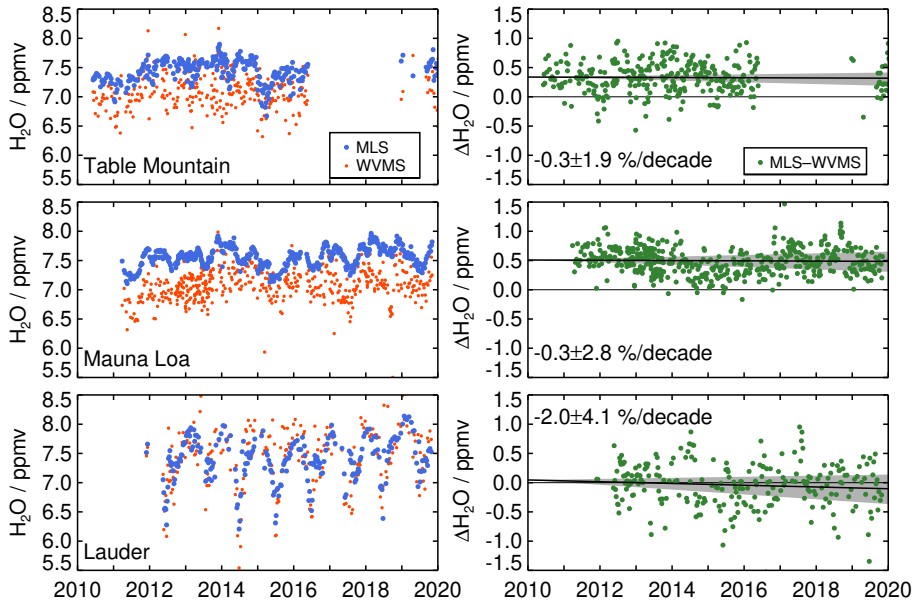

**Figure 4.** Left hand panels show WVMS observations of 55 km lower mesospheric water vapor in red, along with coincident (within ±2° latitude, ±30° longitude) MLS observations in blue. The WVMS data are from roughly week-long integrations, and the MLS points are averages of all spatially coincident observations in a one-week window centered on the middle of each WVMS observation period. The right-hand panels show the corresponding differences (MLS minus WVMS), along with an associated linear fit (with shading indicating the 1-year block bootstrap 95% confidence range for the slope). Drift rates (converted from ppmv to percent) and an associated $2\sigma$ uncertainty are quoted in each panel.

a drift. However, the large gap in the timeseries likely precludes robust interpretation of this comparison. Comparisons at other altitude regions (not shown) give similar results.

### 2.1.4 Inferences from thermodynamic and microphysical studies

Additional confirmation of a probable drift in the lower-stratospheric MLS v4 water vapor observations comes from a micro-
5  physical study building on the work of Lambert and Santee (2018), who examined MLS observations of polar winter lower stratospheric nitric acid ($HNO_3$) and water vapor, along with coincident polar stratospheric cloud (PSC) observations from the Cloud-Aerosol Lidar with Orthogonal Polarization (CALIOP) instrument on the NASA/CNES Cloud-Aerosol Lidar and Infrared Pathfinder Satellite Observation (CALIPSO) mission. MLS and CALIOP data were used, in conjunction with a microphysical model of the equilibrium thermodynamics of supercooled ternary solutions (STSs) and ice clouds, to derive an
10  independent measure of atmospheric temperature in the vicinity of PSCs. These estimated temperatures were then used to quantify the accuracy and precision of various reanalysis stratospheric temperature datasets. The left-hand panel of Figure 5 shows timeseries of average differences between this temperature estimate and temperature from the European Center for Medium Range Weather Forecasts (ECMWF) Interim Reanalysis (ERA-I), interpolated to the same locations, for 2008–2018.



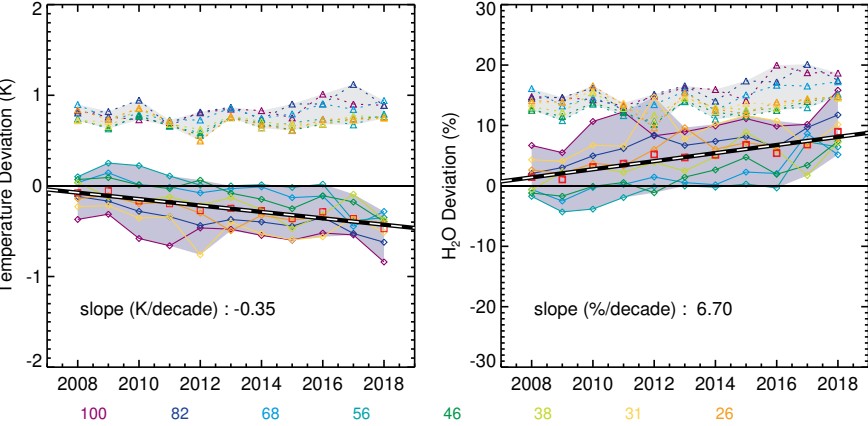

**Figure 5.** Left: MLS $H_2O$ is used to calculate the water-ice frost point (in the presence of CALIOP ice PSCs) for observations south of 60°S, between 100 and 26 hPa and days 140 and 230 of each year. The annually and spatially averaged difference between the interpolated ERA-I temperature and the MLS-calculated frost point is shown by the solid lines with diamonds. Colors indicate the MLS pressure levels (hPa; see legend). Shading encompasses the range of deviations for all pressure levels considered, and the red squares are the means over all pressure levels. Triangles connected with dotted lines show the standard deviations of the individual differences (with shading again indicating the range of these values). The right-hand panel shows the equivalent differences and standard deviations if the ERA-I temperatures are taken to be unbiased and stable and the same microphysical calculations are instead used to provide an estimate of water vapor. This is then compared to MLS water vapor observations (MLS minus estimates from microphysics).

Colors indicate different pressure levels as indicated in the legend. MLS and CALIOP measurements before 2008 and after 2018 were not sufficiently colocated to be included in this analysis. The implied drift in reanalysis temperature, assuming no drifts in the MLS and CALIOP data, is estimated to be –0.35±0.04 K/decade. Only rarely does the Arctic winter polar vortex get cold enough to form large-scale water-ice PSCs, so this analysis is limited to the Antarctic.

5    Conversely, this analysis can be recast into one that provides insights into the MLS water vapor drift by instead assuming that the ERA-I temperatures are unbiased and, more critically, stable over the timeframe considered. The latter assumption is quite reasonable given that, over this period, a consistent portfolio of observations were used in the ERA-I reanalysis (e.g. Long et al., 2017). The microphysical framework described above can then be used to derive an independent estimate of stratospheric water vapor in Antarctic regions containing water-ice PSCs. The right-hand panel of Figure 5 shows the relationship between

10    water vapor estimated using this approach and MLS v4 $H_2O$. This relationship exhibits a 6.7±0.8%/decade drift, with MLS v4 again moistening. In this case the uncertainty corresponds to a simple $1\sigma$ estimate from a linear least-squares fit (i.e., with no bootstrapping).



### 2.1.5 Water vapor summary

We have found strong evidence for a drift in the MLS v4 water vapor dataset starting around 2010. Balloon measurements indicate statistically significant drifts as large as 11%/decade in the lower stratosphere. Comparisons with ACE-FTS show a smaller 2–3%/decade drift over much of the vertical range, increasing to as much as ~7%/decade in the lower stratosphere. Consideration of PSC microphysics in the Antarctic corroborates such a drift, and comparisons with ground-based microwave observations of lower mesospheric water, while not showing any statistically significant drift, are largely not inconsistent with a ~2%/decade drift. Comparisons with SABER lack sufficient statistical significance to comment on the drifts. We also note that Randel and Park (2019), who examined the time-lagged relationships between tropical tropopause cold-point temperature and lower stratospheric humidity, also found evidence for a positive drift in MLS v4 water vapor, peaking at 7–9%/decade in the northern hemisphere lower stratosphere, consistent with the findings above.

### 2.2 Ozone measured by the MLS 190-GHz receiver

Ozone has many strong lines throughout the microwave spectrum, with multiple lines in each of the spectral regions observed by MLS. The MLS ozone "standard product" is retrieved from radiances around 240 GHz, primarily selected because this region includes the strongest lines, providing information over a broader altitude range than is available from the other receivers, spanning from the upper troposphere to the upper mesosphere. Ozone profiles are also retrieved from each of the other four MLS spectral regions, including from strong signals in the 190-GHz and 640-GHz receivers. Ozone signals at 118 GHz and 2.5 THz are noisier and, in the latter case, time limited, so they are not considered here. The MLS 190- and 640-GHz ozone data products can be found in the "L2GP-DGG" data files.

Initial validation of MLS stratospheric ozone focused on the version 2 data set (Jiang et al., 2007; Froidevaux et al., 2008), and average differences between that version and subsequent versions have been small, apart from some vertical oscillations (mainly at low latitudes), which have been reduced in the v4 data compared to previous versions (see Livesey et al., 2020). As noted above, the standard MLS stratospheric ozone product has been shown to be "very stable" in comparison to long-term coincident profiles from ozonesondes and lidars (Hubert et al., 2016); specifically, MLS stratospheric ozone exhibits small or negligible drifts (mostly under 1–2%/decade) relative to those "ground-truth" networks.

The top panel of Figure 6 shows an example of monthly zonal mean ozone time series (2005 through 2019) from the MLS 190-, 240-, and 640-GHz spectral bands in v4. These MLS observations are aggregated from monthly sets of typically hundreds of daily (day and night) colocated measurements in different latitude bins, and the resulting monthly zonal mean variations measured by the different MLS bands track each other very well. A strong Quasi-Biennial Oscillation (QBO) signal is seen when anomaly timeseries are computed (middle panel), with the ozone measurements from the different MLS bands again tracking each other well. Taking differences between the anomalies of the individual ozone products (bottom panel) removes essentially all of this interannual variability, leaving signals with small relative variability, from which simple linear fits indicate robust drifts in some of the inter-product differences. A drift in the 190-GHz time series seen (in blue) in the top two panels is highlighted by the inter-product differences shown in the bottom panel.

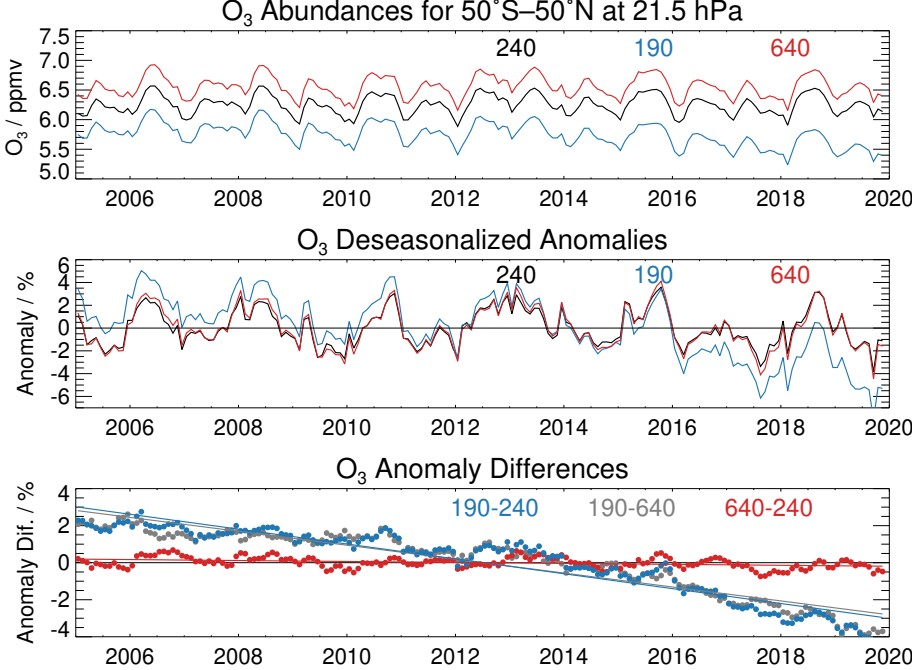

**Figure 6.** The top panel shows a timeseries of MLS monthly mean ozone measurements from the 190-, 240-, and 640-GHz receivers at 21.5 hPa between 50°S and 50°N. The middle panel shows the same timeseries as deseasonalized anomalies (in percent), and the bottom panel shows timeseries of differences between those anomalies for the various combinations of receiver pairs, along with simple linear fits.

Profiles of the resulting drifts between MLS ozone time series from the different bands are displayed in Figure 7, for different time periods for 50°S–50°N. Drifts at higher latitudes (poleward of 50°) or in narrower latitude ranges (not shown) are generally consistent with those in Figure 7. These comparisons conclusively indicate that, in the MLS v4 record, the 190-GHz ozone measurements are drifting compared to both the 240-GHz and the 640-GHz ozone measurements, whereas the latter two

5  show essentially no drift in their differences. As with earlier results, error bars shown are $2\sigma$ bounds, from a block-bootstrap resampling (with replacement, using year-long blocks). Both Figure 6 and Figure 7 also show larger drifts in 190-GHz ozone in more recent years. Results from the first period (2005–2009, upper panel) show very little drift (within the error bars), while the full time period (middle panel) and especially the 2010–2019 period (bottom panel) show worsening drifts for the 190-GHz ozone product versus the other two MLS v4 ozone products. At 22 hPa and smaller pressures (higher altitudes), the

10  190-GHz ozone product exhibits statistically significant drifts with respect to the 240- and 640-GHz results, with values of about –3%/decade for the overall period (2005–2019); most of the drift occurs after 2010, at a rate of about –4%/decade. Drifts over the 100–22 hPa pressure range are less consistent in sign, though still statistically significant in some cases. However, the low abundances of ozone at these levels make reliable assessment of fractional drifts challenging.

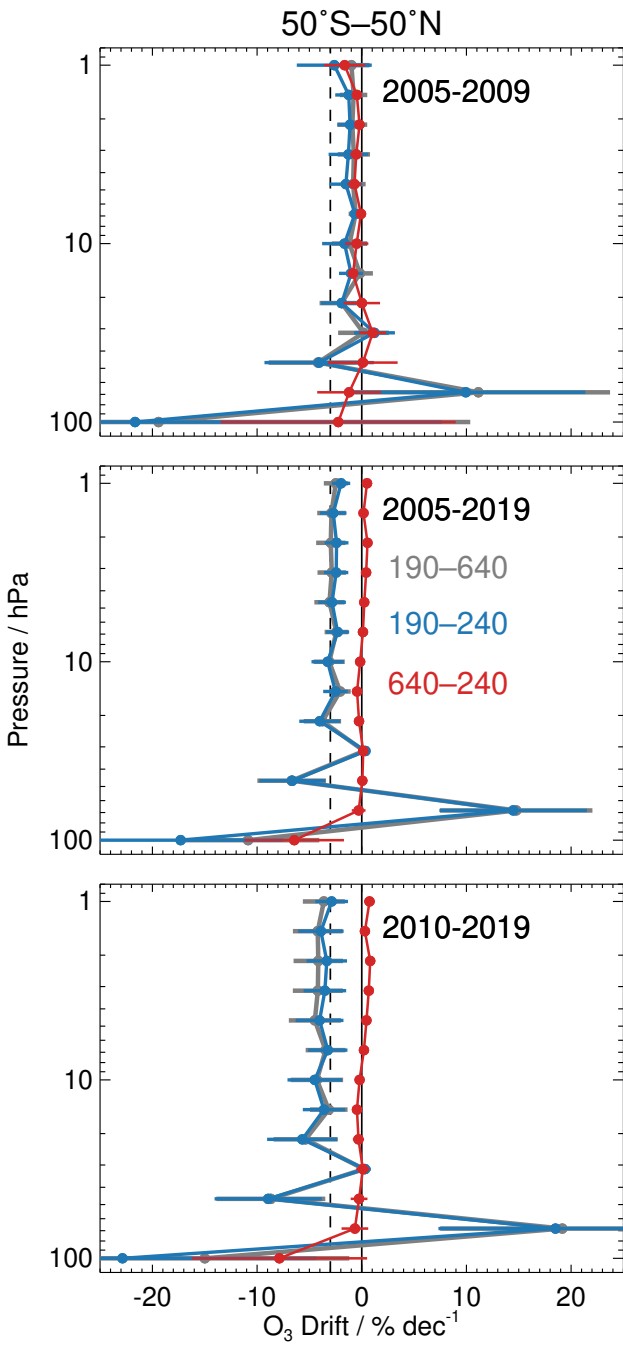

**Figure 7.** Profiles of linear fits obtained from differences between the MLS v4 ozone products based on the 190-, 240-, and 640-GHz receivers over 50°S–50°N for different time periods (see legend). Error bars are $2\sigma$ intervals estimated from block bootstrapping. The solid vertical line marks zero drift and the dashed vertical line indicates a –3%/decade drift.

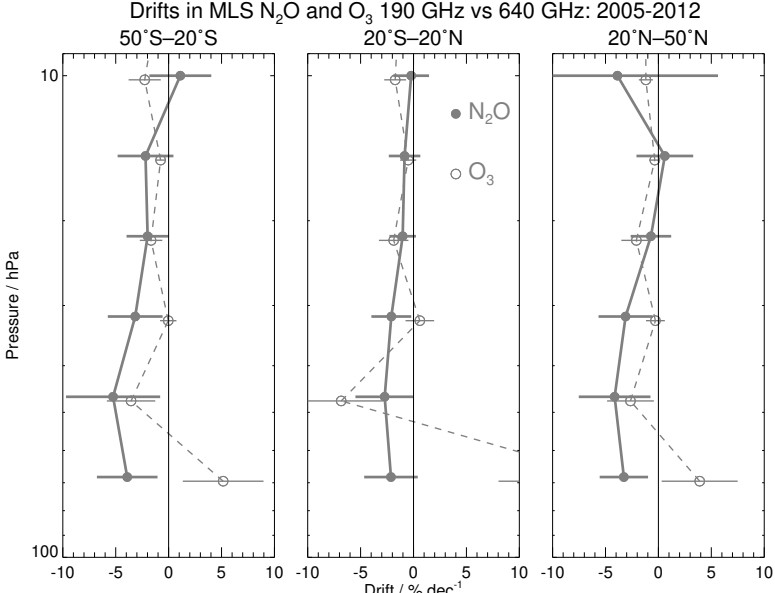

**Figure 8.** Drifts between MLS 190- and 640-GHz $N_2O$ products from 2005–2012 over three latitude bins. Also shown (dashed line) is the corresponding drift seen between MLS 190- and 640-GHz ozone. Error bars are $2\sigma$ intervals estimated from block bootstrapping, as described in the text.

## 2.3 $N_2O$ from MLS 190- and 640-GHz measurements

In MLS Level 2 versions 3 and earlier, the standard product for $N_2O$ was obtained from the 640-GHz receiver. However, this measurement had to be discontinued in June 2013 because of rapid degradation in the $N_2O$-specific elements of the 640-GHz signal chain resulting from a component failure. In order to provide a consistent $N_2O$ record for the entire MLS observation

5  period, the $N_2O$ standard product for v4 (and later versions) is instead obtained from 190-GHz radiances. Characteristics of the two MLS $N_2O$ products and their differences are discussed in the MLS data quality document (Livesey et al., 2020). Although agreement is good overall, the v4 190-GHz $N_2O$ product has a moderate high bias at 68 hPa and a strong high bias at 100 hPa. Consequently, the 100-hPa 190-GHz $N_2O$ data are not recommended for scientific use in v4.

The v4 190-GHz $N_2O$ retrievals have drifted with respect to the 640-GHz $N_2O$ data, as shown in an analysis of the eight

10  years of available 640-GHz data from 2005 through 2012 (Figure 8). These results are obtained in the same manner as those for ozone discussed above but for just the two $N_2O$ bands through 2012. Although the values depend slightly on latitude, in general there are relative drifts from –3 to –5%/decade near 46 to 68 hPa, diminishing to near-zero drift at 10 to 15 hPa (with no clear drift seen in the upper stratosphere, not shown). For comparison, the ozone drifts for the same radiometers (190- versus 640-GHz) are shown for the same period (dashed lines). Between 46 and 10 hPa, the 190-GHz ozone and $N_2O$ drifts agree

15  within their respective error bars.



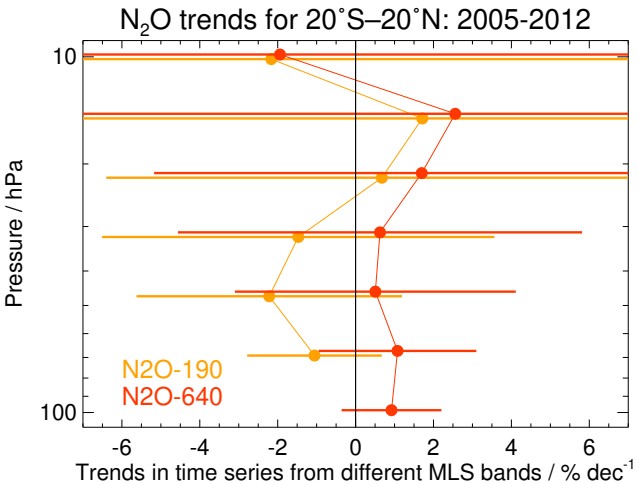

**Figure 9.** 2005–2012 trend seen in MLS 190- and 640-GHz tropical (20°S–20°N) $N_2O$. Error bars are $2\sigma$ intervals estimated from block bootstrapping.

Surface observations show that tropospheric abundances of $N_2O$ are increasing by 2.6%/decade (World Meteorological Organization, 2014), and models predict that this results in an increasing trend in stratospheric $N_2O$ abundances. As shown by Froidevaux et al. (2019), MLS v4 tropical lower stratospheric 640-GHz $N_2O$ displays a positive trend of order 1%/decade for 2005–2012, with error bars encompassing the range from zero up to more than 2%/decade (Figure 9), consistent with (although

5   on the low side of) tropospheric $N_2O$ trends. In contrast, the tropical lower stratospheric MLS v4 190-GHz $N_2O$ trends have values of –1 to –2%/decade. Stronger negative tendencies in v4 190-GHz $N_2O$ are seen in the years since 2014 (not shown, see Froidevaux et al., 2019).

Figure 10 presents further confirmation of a drift in the MLS v4 190-GHz $N_2O$ product from comparisons with the ACE-FTS record, employing the same ±1° latitude, ±8° longitude and ±12 hour coincidence criteria as Figure 2. These comparisons

10   show the 190-GHz $N_2O$ drift having a more pronounced vertical gradient than is seen in $H_2O$ and $O_3$, with no significant drift in the mid-stratosphere and a very strong, and statistically significant, drift of around –15%/decade (±~10%/decade) during 2010–2019 in the lower stratosphere. For comparison, preliminary studies (not shown) quantifying 2004–2020 drifts between the ACE-FTS $N_2O$ record and the newly released "version v3.0" $N_2O$ dataset from the Odin Submillimeter Radiometer (SMR, Murtagh et al., 2002) indicate a ~3–4%/decade drift in the lower stratosphere, with ACE-FTS increasing.

15   In summary, the MLS v4 190-GHz $N_2O$ product shows a significant negative drift, with –3%/decade seen in comparisons with the MLS 640-GHz observations from 2005–2012. These comparisons also indicate a faster drift at higher latitudes (not shown), as mentioned in Strahan and Douglass (2018). Drifts as large as –15%/decade over 2010–2019 are seen in the lower stratosphere compared to ACE-FTS, with even larger drifts seen at 100 hPa, a level that is not recommended for scientific use in the MLS v4 dataset.



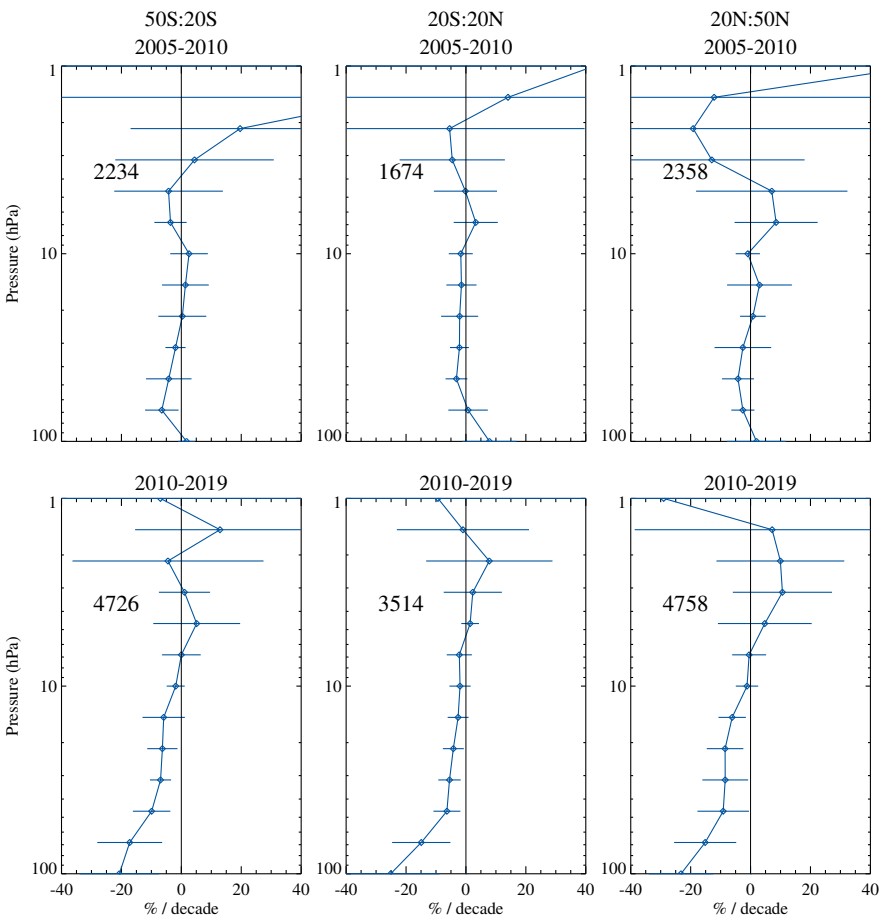

**Figure 10.** As Figure 2, but comparing the MLS 190-GHz $N_2O$ product with the $N_2O$ measurements from ACE-FTS.

## 3 Insights into causes of the drifts, and pathways to their partial amelioration

### 3.1 A drift in 190-GHz "sideband fraction"

The MLS instrument observes thermal microwave emission from the Earth's limb. The radiance signals from which all the MLS products discussed in this paper are derived are received by a 1.6-m primary "GHz" antenna. This antenna scans the limb vertically from the surface to ~90 km altitude every ~24.6 s, looking in the forward direction from the Aura spacecraft (a separate "THz" antenna is used for measuring the OH radical). The microwave signals are then passed through a series of splitters that subdivide the signals into four spectral regions: 118 GHz (for which there are two independent receivers), 190 GHz, 240 GHz, and 640 GHz. All of the MLS receivers employ the heterodyne detection approach, whereby atmospheric signals are mixed with a signal from a Local Oscillator (LO) in a non-linear signal element such as a diode. This non-linear mixing results in Intermediate Frequency (IF) signals that effectively correspond to the atmospheric spectrum in the region





**Table 1.** The key spectral lines in the MLS 190-GHz band ($f_{LO} = 191.9\,\text{GHz}$).

| Molecule | Frequency / GHz | Sideband | Drift direction |
|----------|-----------------|----------|-----------------|
| $H_2O$ | 183.3 GHz | Lower | Increasing |
| $O_3$ | 206.1 GHz | Upper | Decreasing |
| $N_2O$ | 201.0 GHz | Upper | Decreasing |

near the LO frequency, $f_{LO}$, but shifted down in frequency by an amount equal to $f_{LO}$. The parts of the original spectra below $f_{LO}$ are shifted to negative IF space, essentially "folding" them on top of the positive IF signals that correspond to the original spectra above $f_{LO}$. The IF signal is thus a superposition of the original signals above and below $f_{LO}$. Mathematically speaking:

$$S_{IF}(f_{IF}) = \alpha(f_{IF})\,S_{atm}(f_{LO} - f_{IF}) + \beta(f_{IF})\,S_{atm}(f_{LO} + f_{IF}), \tag{1}$$

where $S_{atm}(f)$ is the atmospheric spectrum (e.g., between 176 and 208 GHz for the MLS "190-GHz" signals, for which $f_{LO} = 191.9\,\text{GHz}$), and $S_{IF}(f_{IF})$ is the resulting IF spectrum (between 0 and ~16 GHz for the 190-GHz example). The terms $\alpha$ and $\beta$ are IF-dependent "sideband fractions" describing the contribution to the IF signal from atmospheric signals below and above the LO frequency, respectively. In most cases these sideband fractions are around 0.5, while for the 118-GHz receiver, the upper sideband signals are blocked by a waveguide filter, giving $\alpha \simeq 1$, $\beta \simeq 0$. In practice, factors related to the MLS

calibration scheme and non-neligible thermal emission from the primary antenna and other optics, dictate that the sum of $\alpha$ and $\beta$ typically falls slightly short of unity, but that has no consequence for the issues discussed here.

Figure 11 shows illustrative simulated atmospheric spectra and intermediate frequency spectra for the 190-GHz signals that are central to the MLS products affected by the drift described above. The spectral lines that give rise to the MLS 190-GHz products are listed in Table 1, which also summarizes the drift direction for each product. As summarized in the table, water

vapor (in the lower sideband) shows an increasing drift, while ozone and $N_2O$ (upper sideband) exhibit a decreasing drift compared to MLS observations in other spectral regions and to other data records. This strongly implies that a slow change in sideband fractions, with the lower sideband being increasingly favored, is at least partly responsible for the observed behavior.

### 3.2   Other MLS 190-GHz data products

The standard product for $HNO_3$ uses radiances from lines in the 190-GHz region for pressures smaller than 22 hPa (with

240-GHz information used at larger pressures). Much of the 190-GHz $HNO_3$ information derives from a cluster of lines in the lower sideband. Accordingly, as with $H_2O$, an increasing drift should be expected in the MLS mid- and upper stratospheric $HNO_3$ product. However, comparisons with ACE-FTS (not shown) indicate no statistically significant drifts in the mid- and upper-stratosphere. This likely reflects the poor signal to noise for MLS $HNO_3$ measurements at these altitudes, given the small abundances of $HNO_3$. Additionally, $HNO_3$ has a plethora of lines in the microwave spectrum, including many in the

190-GHz upper sideband. This likely further dilutes the impact of drifts in sideband fraction. Further to this point, we note that



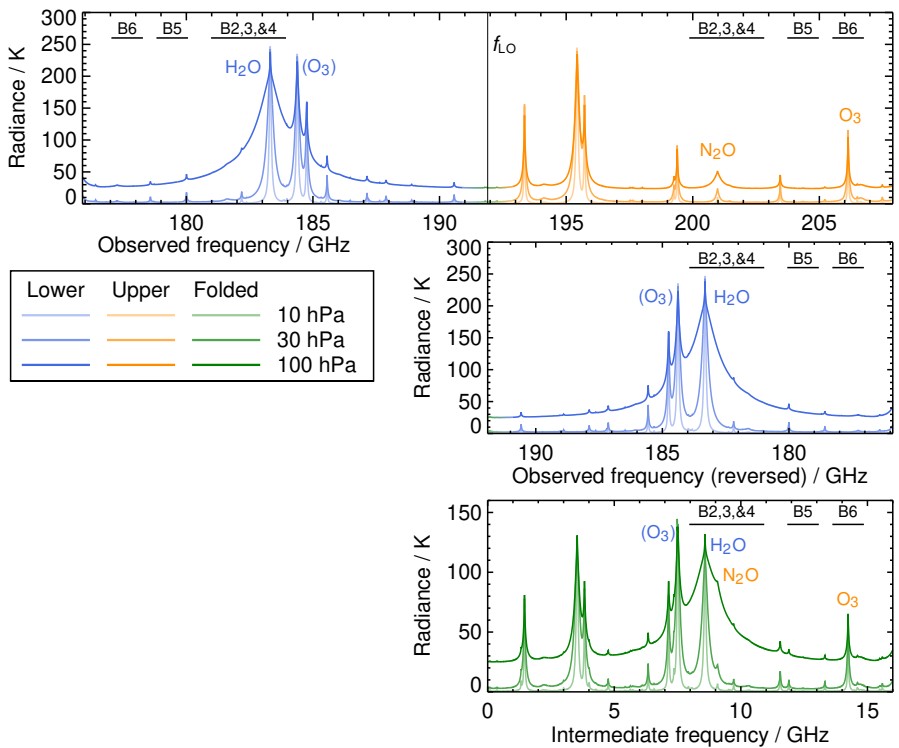

**Figure 11.** Illustration of the 190-GHz spectral region observed by Aura MLS and of the "sideband folding" heterodyne down-conversion process. The top row shows the atmospheric spectrum observed by MLS on either side of the Local Oscillator at $f_{LO}$ =191.9 GHz for tangent points at 10, 30, and 100 hPa (pale to dark colors). The second row shows the "lower sideband" ($f < f_{LO}$) signal folded about $f_{LO}$ (note the reversed x-axis). The bottom plot shows the Intermediate Frequency (0–16 GHz) signal corresponding to a 50/50 sum of the lower and upper sideband signals. The locations of MLS spectral bands 2–6 are specified (bands 2, 3, and 4 overlap significantly and are shown with a single line). The key spectral lines for water vapor, ozone, and N₂O are identified, with color denoting the sideband in which the line is found. Note that the two ozone lines around 184 GHz (labeled parenthetically) are not covered by any MLS spectral bands. The bulk of the ozone information from the MLS 190-GHz spectral region instead derives from the 206-GHz line in the upper sideband covered by band 6. MLS bands 4 and 5 measure HNO₃ and ClO, respectively, whose signals are too small to be seen clearly on this plot.



no consistent drift signals are seen in comparisons between the MLS measurements of $HNO_3$ at 190, 240, and 640 GHz (not shown).

The MLS HCN product derives from a cluster of lines around 177.3 GHz, in the 190-GHz lower sideband. While an increasing drift is therefore expected in this product, comparisons with ACE-FTS (not shown) instead indicate a statistically

significant decreasing drift in the lower stratosphere, ranging over 10–15%/decade at 100 hPa, decreasing to 5%/decade at 46 hPa. However, the MLS HCN product is known to be subject to systematic errors at these levels, with 22 hPa being the largest pressure (lowest vertical level) at which this product is recommended for scientific use (Livesey et al., 2020). These errors likely originate from the significant overlap at these altitudes between the lower-sideband HCN signals and a strong ozone line at 206.1 GHz, in the 190-GHz upper-sideband. (That ozone line, targeted by MLS band 6, provides the bulk of

the information for the 190-GHz ozone data product). This overlap, combined with the drift in the retrieved 190-GHz ozone product, likely accounts for the unexpected sign of the HCN drift.

### 3.3    Quantifying sideband fraction

Although the MLS sideband fractions were measured as part of the pre-launch MLS calibration (Jarnot et al., 2006), investigation has shown that (in contrast with those of the other MLS receivers) the 190-GHz post-launch sideband fractions are not

only different from the pre-launch-measured values, but are also slowly drifting. This has been found through careful analysis of the in-orbit MLS radiance observations in specific channels over limited altitude ranges, where the observed signal is a combination of a large optically thick radiance in one sideband and a small optically thin radiance in the other. More precisely, when viewed at limb tangent altitudes around 30 hPa, the region in the center of the 183-GHz $H_2O$ line (located in the lower sideband) is optically thick (due to the strong absorption by water vapor); signals from this line are combined with those in the

optically thin spectral region in the upper sideband. Since it is optically thick, the signal in the saturated line-center region of the lower sideband is only weakly affected by the abundance of water vapor (or any other species) and is essentially a measure of temperature in the region of the upper stratosphere where limb rays first become optically thick at these frequencies. This signal is multiplied by the lower sideband fraction, which effectively acts as a scaling factor for the atmospheric signal. The upper sideband signal, by contrast, is very small at these altitudes and is largely determined by dry air and water vapor con-

tinua. Given knowledge of the atmospheric temperature profile (e.g., from MLS radiance observations in other spectral regions, and/or from meteorological analysis fields), the expected radiance signal in both sidebands can be readily estimated using a "forward model" radiance calculation and compared to the observed radiances. Such forward model calculations are routinely performed as part of the MLS Level 2 processing.

Figure 12 shows a timeseries of such radiance differences (observed minus calculated, i.e., the difference between the

measured radiances and those predicted from forward model calculations and pre-launch measurements of $\alpha$ and $\beta$ according to equation 1), along with a comparable calculation for the channels in the center of the 235-GHz $O_3$ line measured with the MLS 240-GHz receiver. The 183-GHz radiance differences have a baseline of ~8 K after launch, show a significant ~0.7 K drop in late 2004, and then a comparable increase in early 2006. Those changes are followed by a slow increasing trend from ~8 K in early 2006 to ~10 K in late 2019 (with a strong increase in the late-2018 timeframe). In contrast, 235-GHz radiance





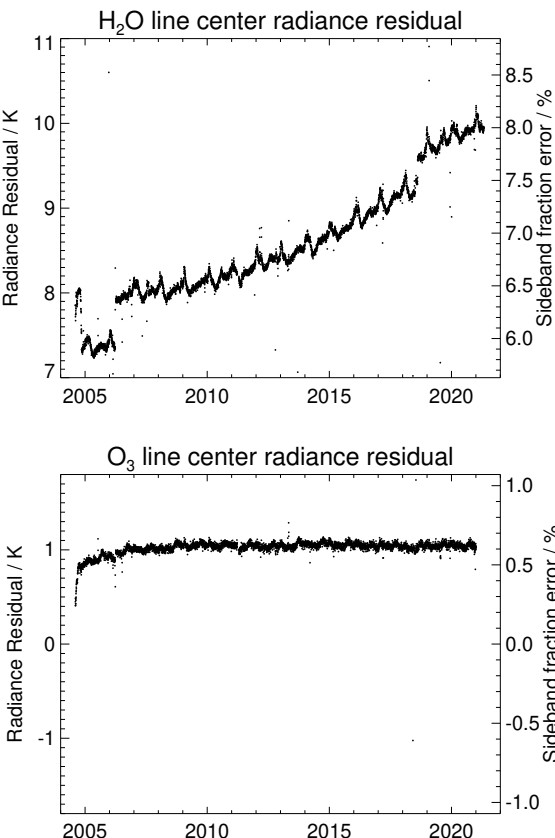

**Figure 12.** Difference between radiances observed in the center of the water vapor (upper panel, 183 GHz) and ozone (lower panel, 235 GHz) lines and those predicted using the MLS forward model and retrieved temperature, etc. for limb tangents around 31 hPa. The offset and temporal variability in the water vapor case is a measure of changes in the 190-GHz sideband fraction from the pre-launch estimate (approximately shown in the right hand axis).

differences are stable to within ~0.1 K or better, and the absolute offset is only ~1 K, which is reasonable given the expected accuracy of the MLS-derived temperature fields used (which also include information from the NASA Global Modeling and Assimilation Office's "Forward Processing for Instrument Teams" data stream) and other assumptions in this calculation.

Other than a change in sideband fractions, the most plausible alternative factor that could give rise to the post-2010 drifting

5  behavior of the 183-GHz signals shown in Figure 12 would be some time-dependent variation in the MLS temperature dataset (which would need to change by an amount comparable to the ~2 K change in radiance residual). We see no signs of such a drift in comparisons of the MLS temperature product with other datasets. Furthermore, such a temperature drift would affect both the 190- and the 240-GHz calculations in Figure 12 in a similar manner. The right-hand axis in each panel of Figure 12 shows the approximate sideband fraction change (compared to the pre-launch calibration value) that would account for the difference

10  between observed and calculated radiances, obtained by solving equation 1 for $\alpha(f_{IF})$ and $\beta(f_{IF})$ using MLS observations of



$S_{IF}(f_{IF})$ and forward model estimates of $S_{atm}(f_{LO} - f_{IF})$ and $S_{atm}(f_{LO} + f_{IF})$, in conjunction with an estimate of $\alpha(f_{IF}) + \beta(f_{IF})$, a sum that is close to unity, obtained from instrument calibration.

The drop in the 190-GHz sideband fraction seen in late 2004, with a "recovery" in early 2006, is associated with a period when the MLS optical bench was maintained at a different temperature compared to that selected for the rest of the mission.

Similarly, the discontinuity in late 2018 corresponds to a change in an attenuator setting in the MLS band 2 signal chain. The fact that this receiver's sideband fraction appears to be affected by such changes in operating conditions implies that it is not as stable as was intended, and thus it is likely also affected by other parameters. In other words, the observed drift in sideband fractions is likely to be a symptom of aging in some more fundamental receiver parameter that is not directly measured within the MLS instrument. Further, we should not assume that the sideband fraction drift is the only consequence of such aging –

other calibration parameters may be drifting also, in less obviously identifiable ways, contributing to the drifts seen in the MLS 190-GHz observations.

The ~8 K post-launch offset in the 183-GHz radiance differences in the saturated line-center region, corresponding to a ~6.5% change in sideband fraction, is also notable. Again, this is likely to be linked, at least in part, to instrument operating temperature, as ground-based calibration was performed at a different instrument temperature than is experienced in flight. The

fact that no such offset is seen for the 240-GHz receiver again speaks to its greater degree of resilience and stability. Note that the 8 K radiance difference between measured and expected radiances seen in the line center occurs in an altitude region where radiances near the line center are not used in the 190-GHz retrievals, specifically because they do not convey information on water vapor, as mentioned earlier.

Sideband fractions vary across the spectral region observed (e.g., the 190-GHz lower sideband fractions established in

pre-launch calibration vary from ~0.55 in band 2 to ~0.50 in band 6). By extension, we do not expect the drift in sideband fraction to have a constant rate across the 190-GHz bands. The analyses of Figure 12 are only possible in the centers of strong spectral lines, so, in the case of the 190-GHz receiver, we can only obtain information on the fraction at ~8.6 GHz Intermediate Frequency, the region where the water vapor measurements are made. However, some significant degree of covariation with the drift in other spectral regions in the 190-GHz receiver bandpass can be reasonably expected.

Figure 12 implies a ~1.3% drift in the sideband fraction in the decade from 2010 through 2019. Figure 13 illustrates the expected impacts (thick lines) of a 1.3% change in 190-GHz sideband fraction (equivalent to a decade's worth of estimated drift) on the $H_2O$, $N_2O$ and 190-GHz $O_3$ products. These results are scaled from the differences between two retrievals of a full day of simulated MLS observations. For one of these simulations, a perturbation has been applied to the 190-GHz sideband fraction; the other is an unperturbed "control" simulation. These simulations are similar to those discussed by Read et al. (2006)

but updated for v4. Results are expressed in terms of a percentage of the retrieved value (from the control retrieval).

The simulated changes are generally similar to the drifts reported in the comparisons described in earlier sections (thin lines in Figure 13), particularly for water vapor in the middle and upper stratosphere. In the case of 190-GHz $O_3$ and $N_2O$, the simulated sideband fraction change impacts underestimate the magnitude of the drift signatures by around a factor of two. However, the vertical structure of these changes generally agrees with that in the observed drifts in the lower to middle

stratosphere, with a strong vertical gradient in the 190-GHz $N_2O$ impacts.





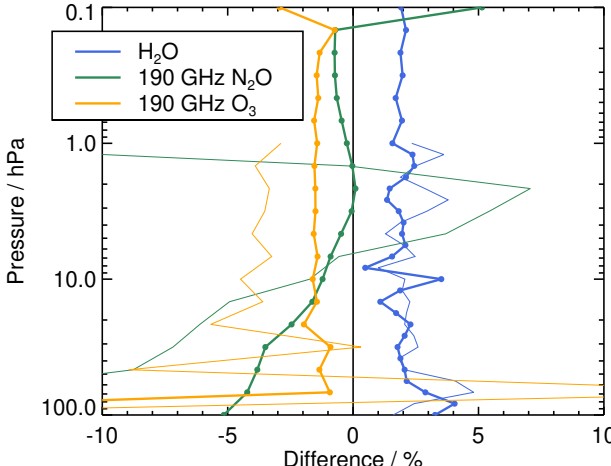

**Figure 13.** Thick lines with circles show the predicted impact of a 1.3% change in 190-GHz sideband fraction on the MLS $H_2O$ (blue), 190-GHz $N_2O$ (green) and 190-GHz $O_3$ (orange) products based on simulated retrievals of a model atmosphere. Thin lines show the corresponding 2010–2019 drift rates. For $H_2O$ and $N_2O$ these are taken from 50°S–50°N ACE-FTS comparisons, while for $O_3$ they are taken from comparisons with the MLS 240-GHz "standard product" ozone (Figure 7).

### 3.4 The MLS version 5 algorithms and dataset

The work to characterize and diagnose the drift in the MLS 190-GHz data products has been the principal focus of the MLS team in developing the algorithms and software for the v5 release of the MLS dataset.

The calculations underlying Figure 12 have been used to generate new time-dependent sideband fraction calibration information (with monthly granularity) throughout the Aura mission. These evolving sideband fractions include not only the slow time drifts, but also the jumps seen earlier in the mission, as well as the ~6.5% initial offset from the values derived from pre-launch calibrations. In the absence of insights into the sideband fraction beyond the 183-GHz line center region used for this diagnosis, the same percentage changes are assumed to apply uniformly across the 190-GHz receiver's spectral range. The analysis of Figure 13 indicates that correction of the sideband fraction drift should go a long way toward improving the drift seen between MLS and ACE-FTS $H_2O$ in the mid- to upper stratosphere, but perhaps not the larger (~7%/decade) drifts seen at lower altitudes in mid-latitudes, nor those seen in comparison to the frost point record.

An additional focus of the v5 development was to alleviate a previously noted (Vömel et al., 2007) significant dry bias in MLS $H_2O$ in the region below the tropopause in v4 and all previous versions. The bias is of order 20% in the tropics, smaller at higher latitudes. The change from the pre-launch value of the 190-GHz sideband fraction increased water vapor in this region by around 10%, somewhat alleviating the bias. Moreover, it was found that, of the various sources of systematic error giving rise to biases in the MLS water vapor observations (Read et al., 2007), another error term with an impact fingerprint that matches that of the observed bias is the vertical pointing offset between the MLS 190-GHz receiver and the nominal boresight pointing (defined as the pointing of "band 8" in the 240-GHz signals used for retrieving temperature and pressure). Adjusting





the vertical pointing offset by 0.0015° (equivalent to a 70 m shift in altitude at the limb) further alleviated this bias (the $2\sigma$ estimated uncertainty in this parameter is 0.003°, based on pre-launch calibration and in-flight observations of the Moon). In addition to improving the $H_2O$ dry bias below the tropopause, these changes had the encouraging benefit of essentially eliminating the v4 bias between the MLS 190-GHz ozone product and the ozone measured by the 240- and 640-GHz receivers.

Another goal of the MLS v5 development was to reduce the clear high bias in the 190-GHz $N_2O$ product at 100 hPa, which rendered it unsuitable for scientific use. Development of a separate retrieval phase (focused on $N_2O$) that changed the way spectral "background" signals were accounted for mitigated that issue. However, it was found that the sideband fraction offset from the pre-launch value implemented as part of the time-varying calibration, as discussed above, significantly increased the $N_2O$ in the lower stratosphere once again. Accordingly, the $N_2O$-focused retrieval phase uses a time-dependent sideband
fraction that reflects only the drift, not the initial offset from pre-launch measurements. Given the proximity of the $N_2O$ line to the region where the analysis in Figure 12 provides insight into sideband fraction, this inconsistency is clearly unsatisfying and warrants further investigation. However, while the absolute value of lower stratospheric $N_2O$ as measured by MLS is subject to some uncertainty, the morphology is expected to be robust.

      As a preliminary assessment of drifts in MLS v5, Figure 14 shows comparisons against ACE-FTS $H_2O$, with MLS v4
results (repeated from Figure 2) in blue and MLS v5 in red. These comparisons indicate that the post-2010 drift in $H_2O$ has been significantly reduced and is no longer statistically significant at $2\sigma$ except in the 20°N to 50°N band at 6.8 hPa (where it exceeds the $2\sigma$ envelope by only ~0.3%/decade). We note that the vertical structure of the v5 and v4 drifts are very similar for all regions and time periods, and that the overall effect of v5 seems to have been to reduce the post-2010 MLS $H_2O$ drift by 2–4%/decade at most pressure levels.

**Table 2.** Comparison of MLS v4 and v5 drifts versus the frostpoint record. All numbers (apart from those for pressure) are % per decade. Differences ($\Delta$) are v4 minus v5.

|          | Lauder |     |     | Hilo |     |     | Boulder |     |     |
|----------|--------|-----|-----|------|-----|-----|---------|-----|-----|
| Pressure | v4     | v5  | $\Delta$ | v4   | v5  | $\Delta$ | v4      | v5  | $\Delta$ |
| 22 hPa   | 8.5    | 5.0 | 3.5 | 3.4  | 0.5 | 2.9 | 9.8     | 8.4 | 1.4 |
| 46 hPa   | 7.5    | 3.8 | 3.7 | 7.1  | 3.5 | 3.6 | 9.1     | 7.5 | 1.6 |
| 68 hPa   | 11.3   | 6.2 | 5.0 | 11.2 | 2.1 | 9.1 | 10.4    | 7.9 | 2.6 |
| 100 hPa  | 4.0    | 2.4 | 1.7 | 2.1  | 0.2 | 1.9 | 6.5     | 6.0 | 0.5 |

Figure 15 reproduces Figure 1 using MLS v5 observations, and Table 2 compares the fitted 2010–2019 drifts between the two MLS versions. The Lauder and Hilo sites show similar reductions in drift at 46 and 22 hPa of around 3%/decade. These sites show larger drift reductions at 68 hPa, as large as 9%/decade for Hilo, while drift reductions at 100 hPa are only ~2%decade. In contrast, the drifts in the Boulder comparisons are diminished less, by 0.5%/decade at 100 hPa and about 1–2%/decade at higher altitudes.



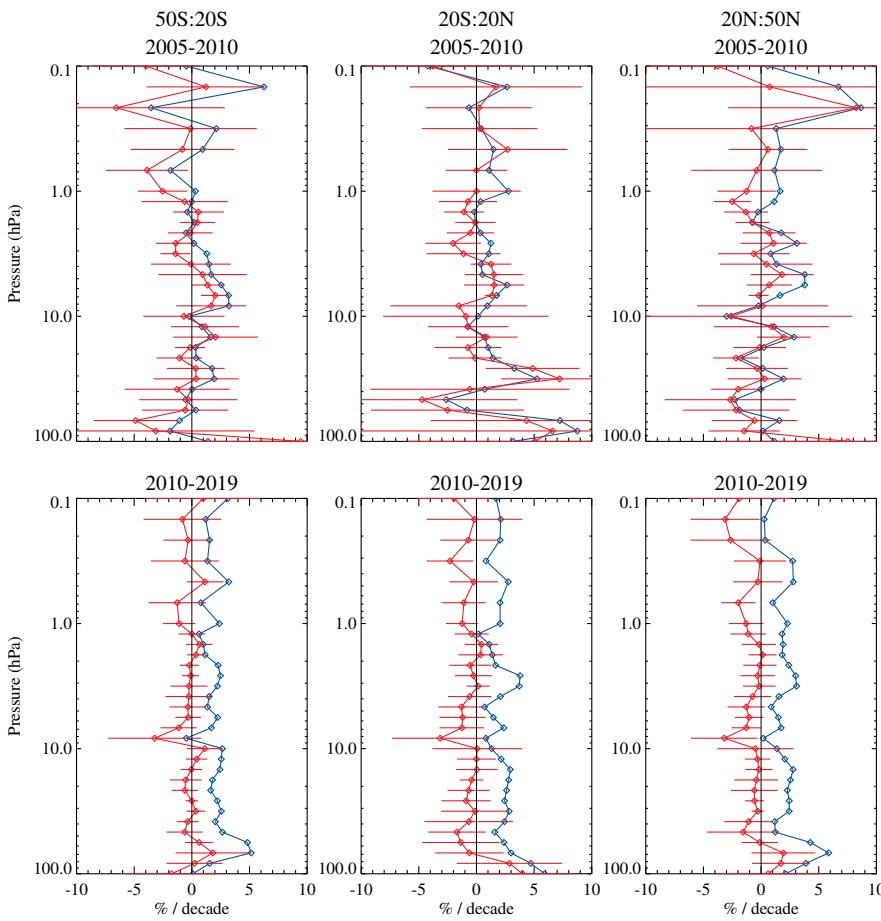

**Figure 14.** Drifts between MLS (v5, red, and v4, blue) and ACE-FTS version 4.1 observations. Coincidence criteria (and v4 results) are as for Figure 2, with error bars on v4 omitted here for clarity.

Broadly speaking, therefore, the drift in water vapor seems to be reduced by about 2–4%/decade over much of the vertical range, though there is some variation in the degree of reduction with altitude and latitude. Such a reduction is sufficient to place the MLS v5 versus ACE-FTS drifts below the level of statistical significance. The frost point comparisons, having indicated stronger drifts in MLS v4 than seen in ACE-FTS comparisons, continue to indicate statistically significant, though reduced,

5   drifts in the MLS v5 water vapor.

The fact that a statistically significant drift remains between MLS v5 water vapor and the frost point measurements, and yet no statistically significant drift is seen in comparisons between MLS v5 and ACE-FTS water vapor, is striking and, thus far, unexplained. This is consistent with the notable disparity in MLS v4 drift estimates derived from the frost point and ACE-FTS records (Figure 1 versus Figure 2). We note historical challenges associated with quantifying water vapor trends from

10   the Boulder record (Kunz et al., 2013) and some debate surrounding the extent to which water vapor trends over Boulder are representative of trends more broadly (Hegglin et al., 2014; Lossow et al., 2018). However, given that, for Figures 1 and 15,





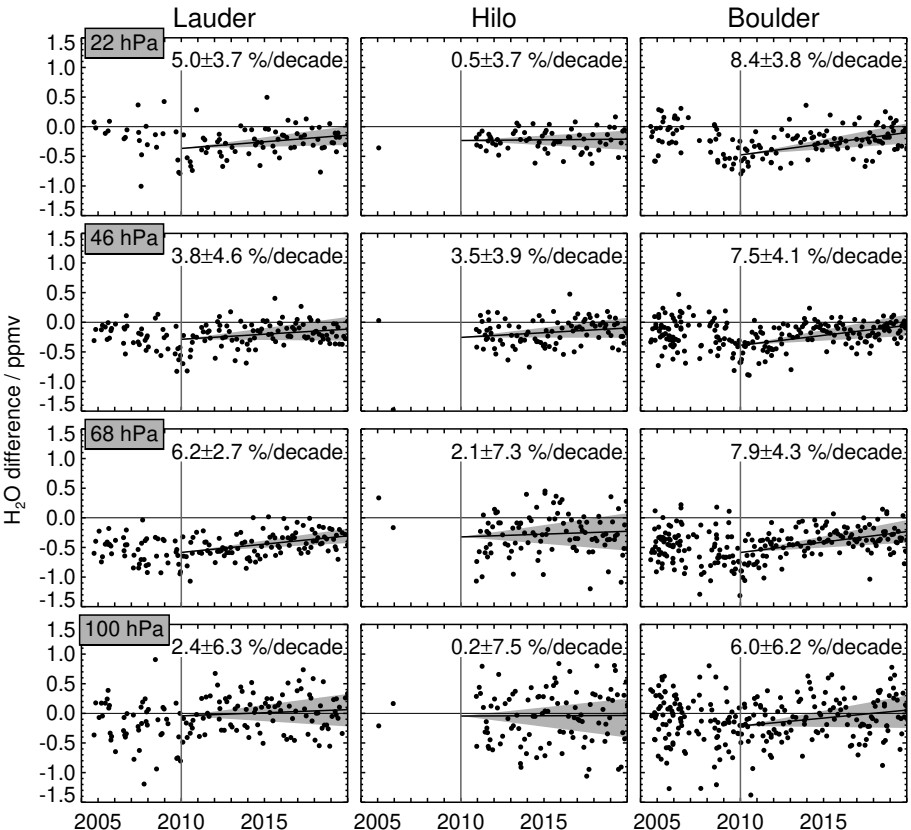

**Figure 15.** As for Figure 1, but comparing MLS v5 observations to frost point measurements.

only MLS profiles in close proximity to the Boulder site are being considered, the issue of Boulder's representativeness should not come into play, as it applies equally to the frost point observations and the MLS observations sampled in that locale, at least to first order.

Figure 16 shows the equivalent comparisons to those in Figure 14 for $N_2O$ (with v4 reproduced from Figure 10). Here, v5 has

5  reduced the magnitude of the drift, which was larger than that for $H_2O$, particularly at lower altitudes. However, a substantial (as large as 10%/decade) and statistically significant decreasing drift remains at pressures of 20 hPa and greater. These figures are consistent with the analysis in Figure 13, which indicates that the expected impact of sideband fraction change, while a good match to the observed drift in water vapor, only accounts for about half of the observed $N_2O$ drift.

## 4   Summary and guidance for users of MLS $H_2O$ and $N_2O$ observations

10  It is clear that the MLS v4 water vapor product is subject to a slow positive drift, likely starting around 2010. A 2–7%/decade drift is seen in comparisons with ACE-FTS v4.1, with the largest drifts between 100 and 46 hPa. Comparisons with balloon-borne frost point hygrometers indicate a larger ~10%/decade drift at these levels. Drifts are also seen in measurements of





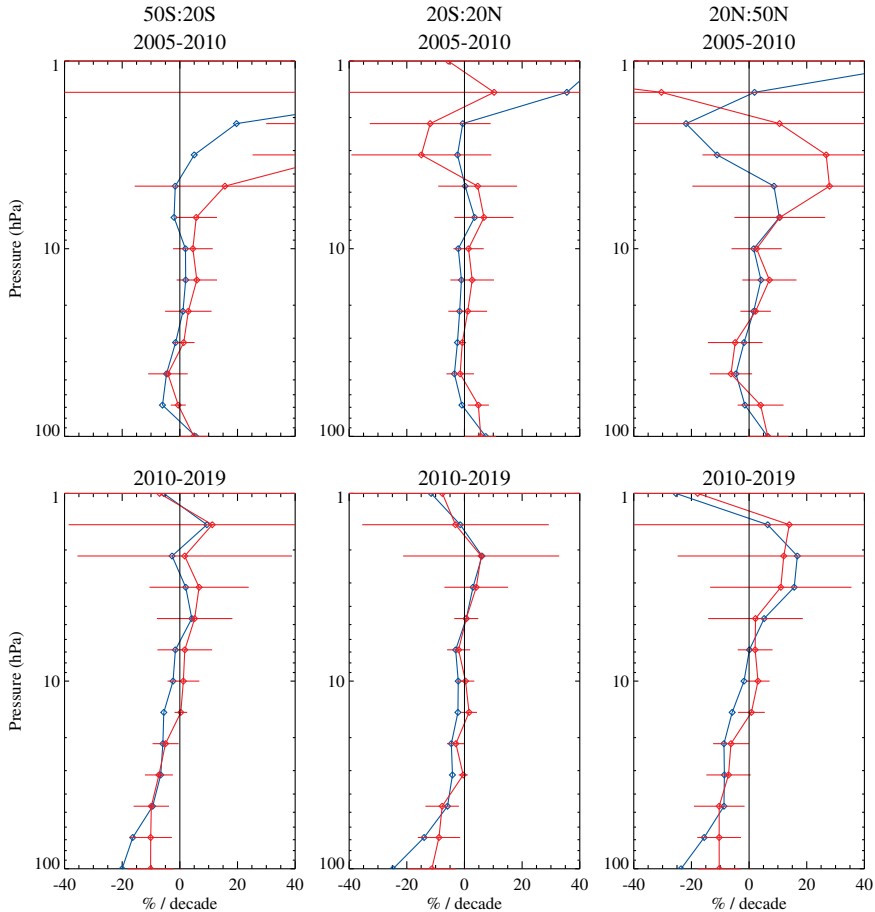

**Figure 16.** As for Figure 14, but for N$_2$O.

N$_2$O and O$_3$ from the same MLS 190-GHz subsystem used to measure H$_2$O. Detailed study of the MLS 190-GHz radiances reveals a drift (and post-launch offset) in the "sideband fractions" of around 1.3% over the last decade that accounts for a significant fraction of the observed drift between MLS and other sensors, with the remainder likely due to other symptoms of an aging receiver. Drifts in 190-GHz sideband fractions will also impact the MLS HCN product and the MLS NNO$_3$ product
5 for pressures smaller than 22 hPa. However, no clear signals of such drifts are seen in comparisons to ACE-FTS.

While the existence of such a drift is clearly unfortunate, we note that it did not start to appear until around six years into MLS on-orbit operations, which corresponds to the original design life of the instrument (now nearly 17 years into a nominal five-year mission). Further, the analyses described above developed to characterize and understand this drift have had the additional benefit of underscoring the stability of the other MLS receivers at 118, 240, and 640 GHz. We emphasize that
10 this drift does not affect the MLS O$_3$, CO, ClO, HCl, HOCl, CH$_3$Cl, CH$_3$CN, CH$_3$OH, BrO, HO$_2$, OH, SO$_2$, temperature, geopotential height, or cloud ice, standard products. We further note that this drift does not call into question the fundamental stability of the microwave limb sounding approach in general.



Data from the entire MLS mission have been reprocessed with the new v5 algorithms, and preliminary comparisons to ACE-FTS v4.1 show a reduction of about 2–4%/decade in the $H_2O$ drifts. As a result no statistically significant drift between MLS v5 $H_2O$ and ACE-FTS observations is seen. Drifts in comparisons of MLS v5 $H_2O$ to frost point measurements are reduced compared to those in v4 but are still statistically significant. For $N_2O$, statistically significant drifts remain between MLS v5

and ACE-FTS, though they are reduced to about half the magnitude of those seen for MLS v4.

Scientists are strongly advised to use MLS v5 data in preference to v4 for all products, but particularly for $H_2O$, $N_2O$, HCN, and $HNO_3$, in light of the work to correct the offsets and drift in the MLS 190-GHz receiver calibration. Users are still advised to apply caution when interpreting the temporal changes in v5 $H_2O$, $N_2O$, HCN, and $HNO_3$ on multi-year time scales, including long-term trends, and to undertake such studies only in close consultation with the MLS team. On the other

hand, studies of spatial and seasonal-to-annual variability in these products, and investigations such as the speed of the "tape recorder" (Mote et al., 1995, 1996) and the impact of the Quasi Biennial Oscillation, should be largely unaffected. The MLS team is planning to continue processing of incoming data with both the v4 and v5 algorithms for the foreseeable future, as continued comparisons between the two versions provide information on the evolution of the drift and its correction.

*Data availability.*  MLS data are available at the NASA GSFC DISC, https://disc.gsfc.nasa.gov. Frost point sonde data can be obtained from

NOAA https://gml.noaa.gov/ozwv/wvap/. ACE-FTS observations are available, following registration, from http://www.ace.uwaterloo.ca, with the data quality information available at https://doi.org/10.5683/SP2/BC4ATC. The SABER water vapor data are available at http://saber.gats-inc.com/, and ground-based microwave observations are archived at https://www-air.larc.nasa.gov/missions/ndacc/data.html.

*Author contributions.*  NJL oversaw the MLS project and wrote much of the text. WGR led the investigation into the MLS 190-GHz drifts and the development of the MLS v5 algorithms. LF led the analysis related to the MLS ozone products and much of the analysis for nitrous oxide.

AL performed the comparisons to ACE-FTS and SABER and led the microphysical studies. MLS contributed to the microphysical studies. MJS and LFM, along with all the aforementioned authors, contributed to the investigation of the drifts and development and testing of the v5 algorithms. RFJ provided insights in his role as MLS instrument scientist. PAW led the implementation of the MLS v5 algorithms and testing of the associated software. DFH contributed data and guidance for the frost point measurements, with KAW and PES similarly contributing for ACE-FTS, and GEN for the ground-based microwave. All authors provided extensive comments and guidance on the manuscript.

*Acknowledgements.*  We thank James Russell III and Pingping Rong of Hampton University for making the SABER $H_2O$ record available and for helpful comments on this manuscript. The research carried out at the Jet Propulsion Laboratory, California Institute of Technology, was performed under a contract with the National Aeronautics and Space Administration (80NM0018D0004). The Atmospheric Chemistry Experiment is a Canadian-led mission mainly supported by the CSA. G. Nedoluha was funded by NASA under the Upper Atmosphere Research Program and by the Office of Naval Research.





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
