# Peer review of "Investigation and amelioration of long-term instrumental drifts in water vapor and nitrous oxide measurements from the Aura Microwave Limb Sounder (MLS) and their implications for studies of variability and trends"

_Atmospheric Chemistry and Physics, 2021_

## Community Comment (CC1)

Comment on "Investigation and amelioration of long-term instrumental drifts in water vapor and nitrous oxide measurements from the Aura Microwave Limb Sounder (MLS) and their implications for studies of variability and trends" by Livesey et al.

We would like to thank the authors for their work that will help to improve the time stability of the Aura MLS observations of $H_2O$ and $N_2O$. We also would like to advertise them that the $N_2O$ drift of the MLS v4 has already been discussed in the evaluation paper of the BASCOE Reanalysis of Aura MLS, version 2 (BRAM2, Errera et al., 2019) using observations from ACE-FTS v3.6, MIPAS IMK v5 and MLS $N_2O$ from the 640 GHz radiometer. The $N_2O$ drift is clearly visible on the Fig. 6 of Errera et al. (reproduced here below) which shows the time series of the monthly mean bias and standard deviation of the differences between BRAM2 and the above mentioned satellite data for several pressure levels and latitude bands. Also, Errera et al. wrote that "analyses of the deseasonalized time series of the biases [of BRAM2] reveal a significant drift of 5, 7 and 5% per decade against ACE-FTS, MIPAS and MLS_N2O_640 for the period 2005–2012 and 10 % against ACE-FTS for 2005–2017" (see on their page 14). We thus believe that Errera et al. could be cited in your paper.

Best regards,

Quentin Errera.

[Figure]

Figure 1: Time series of the monthly mean differences (%) between BRAM2 $N_2O$ and the different observational datasets in the Northern Hemisphere mid-latitudes (30°–60°N) at 46 hPa. The grey shaded area represents the standard deviation of the differences between BRAM2 and ACE-FTS. From Fig. 6 of Errera et al. (2019).

---

## Author Comment (AC1)

**Response to reviews for acp-2021-440**

Nathaniel J. Livesey, William G. Read, Lucien Froidevaux, Alyn Lambert,
Michelle L. Santee, Michael J. Schwartz, Luis F. Millán,
Robert F. Jarnot, Paul A. Wagner, Dale F. Hurst, Kaley A. Walker,
Patrick E. Sheese, Gerald E. Nedoluha

August 23, 2021

We thank the handling editor, Dr. Müller, the two anonymous reviewers, and Dr. Errera for their positive responses to our manuscript, and for their comments, which have improved it. Below we respond to each reviewer in turn, with the reviewer comments quoted in italics.

**RESPONSE TO REVIEW COMMENT FROM DR. QUENTIN ERRERA**

*We would like to thank the authors for their work that will help to improve the time stability of the Aura MLS observations of $H_2O$ and $N_2O$. We also would like to advertise them that the $N_2O$ drift of the MLS v4 has already been discussed in the evaluation paper of the BASCOE Reanalysis of Aura MLS, version 2 (BRAM2, Errera et al., 2019) using observations from ACEFTS v3.6, MIPAS IMK v5 and MLS $N_2O$ from the 640 GHz radiometer. The $N_2O$ drift is clearly visible on the Fig. 6 of Errera et al. (reproduced here below) which shows the time series of the monthly mean bias and standard deviation of the differences between BRAM2 and the above mentioned satellite data for several pressure levels and latitude bands. Also, Errera et al. wrote that "analyses of the deseasonalized time series of the biases [of BRAM2] reveal a significant drift of 5, 7 and 5% per decade against ACE-FTS, MIPAS and MLS_N2O_640 for the period 2005–2012 and 10% against ACE-FTS for 2005–2017" (see on their page 14). We thus believe that Errera et al. could be cited in your paper.*

We apologize to Dr. Errera for not having cited the paper he identifies. This was a particularly unfortunate oversight on our part, given than we have previously discussed this topic with him. We have included a discussion and citation of this paper at the end of the discussion of the drifts in $N_2O$. Specifically, we have added at the end of section 2.3:

> "These results are consistent with those of Errera et al. (2019), who found a drift of 5–7% between MLS and measurements by both ACE-FTS and the Michelson Interferometer for Passive Atmospheric Sounding (MIPAS) on EnviSat (Fischer et al., 2008) during 2005–2012, with 10% drifts seen compared to ACE-FTS for the 2005–2017 period."

**RESPONSE TO ANONYMOUS REVIEWER #1**

*This is a well written paper with a clear motivation, sound methods and result sections, and a recommendation valuable to data users in the conclusion. I agree with the editor that the paper should, strictly spoken, be redirected to AMT since it deals with instrument issues, and not with findings on the atmosphere. However, also in consent with the editor, I find that the paper is of high importance to all MLS data users, for instance in the framework of model validation. For this reason, I concur with the editor that it should better remain in ACP. In this case, however, I also*

*find that a "sexier" title, putting less weight on the instrumental issue and more on the impacts wrt trend estimation would be recommendable. I find the editor's suggestions appropriate, and I'd strongly recommend to change the title for a better fit into ACP.*

*The figures of the paper are clear and useful, and the assessment of the contribution of the sideband fraction to the overall drift is absolutely sound. The short introduction of MLS v5 is concise yet clear. Remaining non-explained differences to water vapor trends measured by balloon-borne frost point hygrometers are openly discussed.*

We appreciate the positive response of the reviewer to this manuscript. See below for a discussion of the title.

*1. Title: see my remarks above (and those of the editor).*

We note that the title was already changed once, in response to the guidance from the handling editor. When initially submitted, the title was:

"Investigation and amelioration of long-term instrumental drifts in water vapor and other 190-GHz measurements from the Aura Microwave Limb Sounder (MLS)"

In response to the editor's suggestion to include a mention of the implications for studies of trends, we revised this to:

"Investigation and amelioration of long-term instrumental drifts in water vapor and nitrous oxide measurements from the Aura Microwave Limb Sounder (MLS) and their implications for studies of variability and trends"

The editor had shorter suggestions, but these omitted mention of the work to ameliorate the drifts, resulting in the new MLS v5 data product. We believe it is important to note this aspect of the work in the title, but are certainly open to further suggestions.

*2. Figure 1 and related text: The description of the fit approach could go a little more into detail. In particular, I miss any explanation how measurement errors were taken into account. Have the individual data points been weighted by their uncertainty within the trend/drift fit? If so, which uncertainty has been used? I am asking since MLS $H_2O$ is retrieved in log space as far as I know. If errors due to measurement noise are more or less constant for all atmospheric situations in log space, this may lead to the conclusion that they can be ignored within the trend fit since they put the same weight on every data point. However, it has to be noted that errors from log(vmr) retrievals scale with the retrieved vmr amount, and, thus, are not equal in vmr space.*

The fit in Figure 1 (and similar analyses) does not consider the reported uncertainties in the MLS (or sonde) observations. This point is already made in the text (page 4, line 26 in the discussion manuscript):

"All differences carry equal weight in the regression (i.e., no accounting is made for estimated precision or other factors)."

We have modified the caption for Figure 1 to further underscore that point, adding the parenthetical text:

"...(with all points weighted equally)..."

Also, the MLS $H_2O$ retrieval is not, in fact, performed in log-space. Rather, the MLS forward model uses a log(vmr) interpolation (in log(pressure)) to populate the points along each ray path (all other species are interpolated linearly in vmr). This representation was chosen over the linear representation used for other species in order to better capture the rapid changes in water vapor mixing ratio with altitude in the upper troposphere.

While the estimated precisions for retrieved $H_2O$ values do vary slightly by value (e.g., at 68 hPa, the precision on a 3 ppmv measurement is ~0.26 ppmv, while 6 ppmv abundances are measured with ~0.32 ppmv precision), this is likely to be driven more by the non-linear nature of the strong water vapor line signature measured by MLS than by the use of the logarithmic interpolation.

*3. P5, l 8/9: "These 2- to 4-periods..."; a reference for this statement should be provided.*

There is no suitable citation for this conjecture, which was provided by co-author Dr. Hurst, the lead for the frostpoint observations. To underscore the speculative nature of this discussion, we have prefixed the sentence with "We believe that...".

*4. Section 2.1.3: As far as I know, the WVMS groups use MLS as a priori in their retrievals. How far can the WVMS measurements be considered independent of the MLS a priori? Some more information about the a priori content of WVMS measurements of the used for validation of MLS would be helpful.*

This point is already discussed in the text, where we state (page 8, line 7–8 in the discussion manuscript):

> "The WVMS retrievals use an MLS-based climatology, which includes seasonal variations (but not interannual variations, thus it is not affected by the MLS drift)."

To make the point clearer, we have reworded to:

> "The WVMS retrievals use an MLS-based climatology, which includes seasonal variations but not interannual variations; thus the WVMS measurements are not affected by the MLS drift."

*5. A remark on Fig. 7: I find the consistency between the ozone measurements at 240 and 640 GHz impressing. Congratulations to this data product!*

We are similarly gratified by this level of agreement.

*6. Table 2 and Fig. 14: For clarity, it should be mentioned in the captions of the figure (and maybe also of the table) that they refer to water vapour (and not $N_2O$) drifts.*

Agreed and enacted. We also took the opportunity to clarify the captions for Figures 1, 3, and 15, which were similarly lacking.

RESPONSE TO ANONYMOUS REVIEWER #2

*The primary goals of this paper have been met and I see no reason to require modification before publication. The authors set out to demonstrate that certain MLS data products derived from observations made with the 190-GHz subsystem had statistically significant long term drifts. Results from comparisons with measurements and model derived results were presented in a systematic fashion that made this very clear. The authors then went on to identify a probable instrumental cause for most of the noticed drifts. The arguments here were clear and concise and informative. After corrections were applied to the processing algorithms, new data versions were produced and a similar set of rigorous comparisons were performed. These comparisons indicate improvements in data quality but did not fully correct for all previously detected "drifts". In particular drifts with respect to the frost point measurements still remain. This does not detract from the results but does leave an open question. Finally the authors aim to give guidance to the scientific community, and this was done concisely at the end of the paper. I suggest no substantive changes to this paper before publication.*
*Publish pretty much as is.*

The reviewer's positive comments are greatly appreciated.

*1. It is mentioned in the abstract and the introduction that comparisons with other sensors have previously indicated drifts. Perhaps it would be good to mention SABER and the WVMS measurements earlier in the paper, before they are introduced in the respective sections.*

We have added discussion of the comparisons with SABER and WVMS to the abstract. For completeness, we have also described the microphysical studies. The new text added to the abstract is:

> "... Microphysical calculations considering the formation of polar stratospheric clouds in the Antarctic winter stratosphere corroborate a drift in MLS v4 water vapor measurements in that region and season. In contrast, comparisons with the Sounding of the Atmosphere using Broadband Emission Radiometery (SABER) instrument on NASA's Thermosphere Ionosphere Mesosphere Energetics and Dynamics (TIMED) mission, and with ground-based Water Vapor Millimeter-wave Spectrometer (WVMS) instruments, do not show statistically significant drifts. However, the uncertainty in these comparisons is large enough to encompass most of the drifts identified in other comparisons..."

The opening words of the following sentence in the abstract were also changed, to avoid repeating "In contrast".

*2. The caption for Figure 14 could mention the results are for $H_2O$.*

Agreed and enacted. We also took the opportunity to clarify the caption for Figures 1, 3, and 15 (and Table 2, per reviewer #1), which were similarly lacking.

*3. I had a handful of even more minor comments but now they don't seem worth mentioning. I may have noticed a missing comma, but it's open for debate so I'll leave it alone.*

Now we're curious, and we are examining all our commas (present or otherwise)!

*This paper is a very nice piece of work and the authors should be proud of how well these very important results were presented.*

This compliment is much appreciated.

ADDITIONAL CHANGES

We identified a shortcoming in the abstract, which contained no quantitative information on the $N_2O$ drift. We have added the following sentence:

> "Comparisons to ACE-FTS and to MLS $N_2O$ observations in a different spectral region, the latter available from 2004–2013, indicate an altitude-dependent drift, growing from 5%/decade or less in the mid-stratosphere to as much as 15%/decade in the lower stratosphere."

Also, for clarity/accuracy, we changed "in 2001" to "after 2001" in the discussion of pressure-induced biases in frostpoint water vapor measurements (page 6, line 1 in the discussion manuscript).